# A Multi-Scale-Enhanced YOLO-V5 Model for Detecting Small Objects in Remote Sensing Image Information

**DOI:** 10.3390/s24134347

**Published:** 2024-07-04

**Authors:** Jing Li, Haochen Sun, Zhiyong Zhang

**Affiliations:** 1Information Engineering College, Henan University of Science and Technology, Luoyang 471023, China; xidianzzy@126.com; 2School of Information Engineering, Henan Mechanical and Electrical Vocational College, Zhengzhou 451191, China; hcsun1994@hotmail.com

**Keywords:** small-object detection, RSI information, YOLO network, residual network, densely connected network

## Abstract

As a typical component of remote sensing signals, remote sensing image (RSI) information plays a strong role in showing macro, dynamic and accurate information on the earth’s surface and environment, which is critical to many application fields. One of the core technologies is the object detection (OD) of RSI signals (RSISs). The majority of existing OD algorithms only consider medium and large objects, regardless of small-object detection, resulting in an unsatisfactory performance in detection precision and the miss rate of small objects. To boost the overall OD performance of RSISs, an improved detection framework, I-YOLO-V5, was proposed for OD in high-altitude RSISs. Firstly, the idea of a residual network is employed to construct a new residual unit to achieve the purpose of improving the network feature extraction. Then, to avoid the gradient fading of the network, densely connected networks are integrated into the structure of the algorithm. Meanwhile, a fourth detection layer is employed in the algorithm structure in order to reduce the deficiency of small-object detection in RSISs in complex environments, and its effectiveness is verified. The experimental results confirm that, compared with existing advanced OD algorithms, the average accuracy of the proposed I-YOLO-V5 is improved by 15.4%, and the miss rate is reduced by 46.8% on the RSOD dataset.

## 1. Introduction

Remote sensing images (RSIs), especially high-altitude RSIs (HRSIs), are mainly generated by satellite or aircraft imaging systems. HRSIs are widely utilized in weapon guidance, emergency rescue, intelligent agriculture, and other fields, and have gradually become the focus of related studies [1,2,3]. However, on account of remote sensing satellites always taking images at high altitudes, the quality of HRSIs is affected by many factors, possibly causing various types of geometric deformation, distortion, blurring, and noise in HRSIs. These factors include atmospheric conditions, illumination variation, and environmental interference, which make it difficult to detect an object in RSIs accurately with a complex background [4,5]. At present, there are still many problems and challenges waiting to be solved in the field of remote sensing object detection (RSOD), including unsatisfactory detection accuracy, detection errors, and missing rate. The existence of these problems not only limits the further application of RSIs, but also causes some problems in practical applications.

With the efforts of researchers at home and abroad, object detection (OD) has made great progress. However, small-object detection (SOD) in complex scenes, especially RSOD, is still full of challenges, especially when there are lighting, occlusion, and similar-background interferences [6,7]. To improve the performance of RSOD, researchers have designed a variety of improved algorithms from different perspectives and problems. In Ref. [8], a new method for RSOD, which combines a feature detection module inspired by a sparse-representation-based generalized Hough transform, was proposed, and it involves the process of learning a dictionary of objects and backgrounds to continuously supplement the sparse image representation of specific classes to realize RSOD.

The existing algorithms for RSOD have made some achievements in dealing with remote sensing objects (RSOs) in relatively simple backgrounds, while they have also encountered a series of problems in complex image backgrounds, such as a low detection accuracy, large detection errors, and a high missing detection rate. In the face of the challenges of a complex environment, there are many limitations in taking the maximum advantage of these algorithms, which hinders their potential to obtain more reliable and accurate RSOD results in practical applications [9,10].

## 2. Related Work

With the advent of high-performance computers and large-scale public datasets, deep learning (DL) has advanced rapidly. To solve the problem of image classification, the deep convolutional neural network (DCNN) was proposed in the literature [11], and breakthrough results have been achieved. With the successful application of DCNNs in image classification, DL-based OD methods have also made significant advancements [12]. Then, DL-based techniques have been swiftly applied to other visual tasks, and have presented better results than conventional methods [13]. Different from the hand-designed feature descriptors used in traditional detection methods, a DCNN generates hierarchical feature representations, from raw pixels to high-level semantic information, and automatically learns from training data, showing a stronger discriminative expression ability in complex contexts. These features, independently learned by the network, have stronger feature representation capabilities [14,15,16].

At present, DL-based OD frameworks are mainly divided into two categories: two-stage detectors and single-stage detectors. The former are typical of region-based convolutional neural networks (R-CNNs), which mainly include Fast R-CNN [17,18,19], Faster R-CNN [20,21,22,23,24], Mask R-CNN [25,26,27,28], etc. As for the latter ones, YOLO-based series detectors are the typical members, such as YOLO and its variants YOLO-V2 to YOLO-V8 [29,30,31,32,33,34,35,36], etc.

A two-stage detection algorithm is a kind of classical detection framework. Its core idea is to carefully classify potential candidate boxes located on the feature map to realize the final detection. Although such algorithms are highly admired due to their high accuracy, their overall detection speed is relatively slow, as the detection process is segmented into two parts, i.e., generation and classification. As a consequence, with the continuous expansion of application scenarios and the increasing requirement for real-time performance, some defects of these algorithms have gradually become prominent and non-negligible. Actually, in real-time application scenarios, detection speed can be a fatal limiting factor. Secondly, due to the need to enumerate a large number of candidate boxes, such algorithms are also expensive in terms of storage and computation. In addition, due to its two-step independence, their model structure is difficult to compress effectively, which further increases the resource overhead. To overcome these shortcomings, the OD algorithm based on a single-stage detector came into being. This kind of algorithm fuses the two steps into an end-to-end network structure to immediately forecast the position and class information of an object through a single stage. This fusion not only simplifies the whole detection process and improves the computational efficiency, but also makes the model easier to be compressed and optimized. Due to the direct regression of the location and category of the object, this kind of algorithm performs well in real time and is widely used in various engineering application scenarios.

As a typical single-stage open-source OD algorithm, YOLO series algorithms have achieved remarkable performances in inference speed and detection accuracy, especially in multi-scale OD, for which YOLO-V5 was first proposed. Hereafter, numerous YOLO-V5-based OD algorithms began to emerge. In Ref. [37], an improved YOLO-V5 method was proposed to detect vehicles in different traffic conditions with a low false detection rate. In Ref. [38], the YOLO V5-IMPROVEMENT model was proposed, increasing the CA attention mechanism, SIoU loss function, etc., to improve the detection efficiency of small objects, thus reducing the rate of misjudgment; the model had not only high accuracy, but also good robustness. In Ref. [39], a novel model was established for colonoscopy polyp detection, which trained the dataset generated by the data augmentation technique, to decreases overfitting and increase the flexibility. In Ref. [40], an optimized YOLO-v5 model, namely D-YOLO-V5, was proposed for the intelligent identification of blade surface cracks. The blade surface crack detection model improved the accuracy of detection to 98.62%.

Although the above related improvements can raise some kinds of performance of the algorithm to a certain degree, there is still room for improvement when facing small targets in complex backgrounds. For example, small objects in remote sensing images (RSSOs) usually lack sufficient contour information, leading to difficulty in distinguishing RSSOs from the background or similar objects, which brings great challenges to the aspect of feature extraction of the OD network. Recently, some new YOLO-based methods have been presented, such as YOLO-V7 and YOLO-V8, and have shown a high accuracy in OD. However, while these improvements increased the number of network layers that can extract rich features of the layers, it will lead to gradient fading of the network, leading to poor training performance of the algorithm. Furthermore, the scale of network parameters, the learning complexity, and the training overhead of these new methods are huge. Meanwhile, different from traditional OD, the detection for RSSOs is prone to being fuzzy, deformed, multi-scale, and so on, which makes OD and recognition much more difficult. Therefore, for better adaptation to small OD tasks in a complex RSI environment, an improved lightweight OD framework, I-YOLO-V5, is proposed and verified in this paper. Our contributions are as follows:(1)To boost the multi-scale feature extraction performance of RSSO, a “Res2Unit”, based on the idea of residual networks, is employed to change the residual unit in the feature extraction network of YOLO-V5.(2)To further avoid network gradient fading and enhance the learning quality, ‘DenseBlock’ in dense connected networks are used to replace the convolutional layer in the detection layer.(3)To strengthen the detection accuracy of RSSOs and decrease the miss rate in complex environments, a fourth scale is employed in the proposed I-YOLO-V5 framework.

The remaining sections of the paper are organized as follows. In Section 2, the framework of YOLO-V5 is introduced in detail. In Section 3, the improvements of the proposed I-YOLO-V5 algorithm in this paper are presented. In Section 4, the simulation results show how the proposed algorithm is effective for RSSO detection. Section 5 presents the conclusions.

## 3. Methodology

YOLO is a single-stage OD algorithm, which is also a typical algorithm. It has good generalization performance, good flexibility, and high efficiency, including the use of anchor boxes to combine classification regression problems with target location problems. Meanwhile, YOLO series algorithms are undoubtedly more suitable for practical engineering applications than two-stage detection algorithms, and their derivative algorithms, in real-time applications.

### 3.1. Introduction to YOLO

The idea of the YOLO algorithm is to divide the input image into a fixed number of grids (like S×S), while each grid predicts a fixed number of bounding boxes and their confidence and class probability, which significantly improves the detection speed of the algorithm to a certain extent. The YOLO algorithm utilizes five values to represent a bounding box, four of which (x,y,w,h) determine the specific position of the bounding box, and one value (*p*) for the confidence of the bounding box, which can be expressed as:(1)$$P\left(Object\right)\times IoU_{pred}^{truth}$$
where $IoU_{pred}^{truth}$ is called Intersection-Over-Union, which can be interpreted as the intersection of prediction and reality, and its value ranges between 0 and 1. It is worth noting that the closer the value of IoU is to 1, the closer the prediction is to reality, and vice versa.

It is not difficult to see that the final output or prediction result of the YOLO algorithm can be quantified as a tensor of S×S×(N×5+C), where S×S represents the number of grid cells after the input image is divided, *N* represents the number of bounding boxes predicted by each grid cell, 5 refers to the five values to determine the bounding boxes, and *C* represents the number of categories of objects predicted by the grid cell.

The framework of the YOLO algorithm is roughly divided into three parts, which are the backbone, the neck, and the head. The backbone part can be considered a DCNN, which employed multi-layer convolution and pooling operations to gradually down-sample the input image to achieve the purpose of extracting high-level features of the input image. Based on the unique role of the backbone, the YOLO family of algorithms utilizes the backbone part with different characteristics to extract the network from the input image to achieve improved algorithm performance, like DarkNet and CSPNet [41].

The neck is composed of convolution layers of different kernel sizes, which plays an important role between the backbone and the head. The task of the neck is to capture the features of different scales from the backbone and combine them to complete the purpose of improving the accuracy of OD.

The head, utilized to predict the class probability and bounding box of the object, is the key part, which consists of a set of fully connected layers that employ the output from the neck to generate the prediction results. It is worth noting that the head also contains a set of anchor boxes with predefined boundary effects to accommodate detection objects of different sizes.

### 3.2. The Principle of YOLO-V5

To improve accuracy, speed, and robustness, the new version of YOLO was proposed to handle some of the earlier limitations and, starting with V3, the YOLO algorithms began to predict bounding boxes at different scales, while extracting features from these scales utilizing the concept of similarity in the feature pyramid network [42].

Figure 1 shows the overall architectural design of YOLO-V5, where the Focus module is designed to achieve better captures of objects at different scales, while the Spatial Pyramid Pooling (SPP) module is utilized to extract features of different scales. The Convolutional Block with Linear Bottleneck (CBL) contains a standard convolution layer, which is mainly used to extract low-level information from the input feature map. The details about specific modules of the YOLO-V5 architecture above is shown in Figure 4.

The loss function of YOLO-V5 is defined as the sum of coordinate prediction error, error and classification error, which can be expressed as:(2)$$\ell =\ell_{box}+\ell_{obj}+\ell_{cls}$$
where $\ell_{box}$ means the regression loss function of the bounding box, $\ell_{obj}$ presents the confidence loss function, and $\ell_{cls}$ is the classification loss function. It is worth noting that each term can define a weight coefficient based on the importance of each part.

In earlier versions, the Intersection over Union (IoU) and Generalized IoU (GIoU) were employed, but they suffered from slow convergence and inaccurate regression. To solve these problems, the CIoU loss function is proposed from three aspects: the overlapping area, center distance, and aspect ratio, which can be defined as:(3)$$\ell_{box}=\ell_{CIoU}=\underset{A}{\underset{︸}{1-IoU}}+\underset{B}{\underset{︸}{\frac{\rho^{2}\left(b,b^{gt}\right)}{c^{2}}+\alpha v}}$$
where part A means the IoU loss, and part B contains the penalty term to measure these factors. More detailed information about the CIoU’s formula can be found in [37].

The confidence loss in Equation (Equation 2) is calculated based on the bounding box confidence score, which is presented in Equation (Equation 1). The confidence score is employed to quantify the likelihood and accuracy of the object in the bounding box. If the object is not in the bounding box, the confidence score takes a value of 0.

The classification loss is calculated for each grid cell, anchor, and category, while the object is detected. These are two methods for calculating the loss; one is the square error of the conditional probability, and the other is the cross-entropy. Binary cross-entropy (BCE), combined with the sigmoid function, is employed, which can be expressed as:(4)$$\;\;\ell_{cls}=\sum_{m=0}^{S^{2}}\sum_{n=0}^{B}\sum_{c\in cls}l_{mn}^{obj}\left[-{\hat{p}}_{m}\left(c\right)\log\left(\sigma \left(p_{m}\left(c\right)\right)\right)-\left(1-{\hat{p}}_{m}\left(c\right)\right)\log\left(1-\sigma \left(p_{m}\left(c\right)\right)\right)\right]$$
where $l_{mn}^{obj}$ means whether the object exists in the n-th anchor box of the m-th grid cell. $p_{m}$ is the predicted category, and ${\hat{p}}_{m}$ represents ground truth category. σ(.) represents the sigmoid function, which can make the loss more numerically stable in terms of value.

### 3.3. The Improved YOLO-V5 Algorithm

To effectively mitigate gradient fading under the premise of ensuring network depth, the ResNet structure is adopted in the algorithm framework of YOLO-V5. The backbone of the original YOLO-V5 algorithm contains four CSP Bottleneck with three convolutions, called C3, where each C3 contains one or more sets of residual units. The specific architecture of the YOLO-V5 algorithm and residual unit are shown in Figure 1 and Figure 2, respectively.

Although the C3 in the structure of YOLO-V5 can effectively increase the depth and receptive field of the network, and improve the ability of feature extraction, the residual unit in C3 reflects multi-scale features by layering, which easily leads to the underutilization of the features of each layer. Meanwhile, it is worth noting that the shortcoming of small object visualization features in remote sensing will further expand the defects of C3 in feature utilization. Based on this, a new residual unit [43] is employed to acquire features to solve this problem. It constructs hierarchical residual class connections in a single residual unit to complete feature extraction, called Res2Unit. Res2Unit enables the representation of multi-scale features at a finer granularity while increasing the range of acceptable domains for each layer to a certain extent. It is not difficult to find that the purpose of increasing the perceptual domain of each layer can be achieved by introducing several small residual terms into the residual unit. The structure of Res2Unit and the basic residual unit are shown in Figure 2.

From the ‘Res2Unit’ in Figure 2, it can be clearly found that the feature map of input is divided into four sub-features on average after the convolutional layer (1×1). The sub-features are named $x_{1},x_{2},x_{3},x_{4}$, respectively, and are the same size. The significant difference between the sub-feature map and the feature map of input is that it contains only 1/4 of the channels. It is worth noting that $y_{1},y_{2},y_{3},y_{4}$ in Figure 2 can be presented as:(5)$$\left\{\begin{array}{c} y_{1}=x_{1}\\ y_{2}=x_{2}\times Conv\\ y_{3}=\left(x_{3}+x_{2}\times Conv\right)\times Conv\\ y_{4}=\left(x_{4}+\left(x_{3}+x_{2}\times Conv\right)\times Conv\right)\times Conv \end{array}\right.$$
where × means convolution, and Conv represents the 3×3 convolutional layer.

As can be found from the neck in the original YOLO-V5 structure, each detection layer contains no less than seven convolutional layers. Although these convolutional layers can fully merge the feature maps of RSSOs from different layers and enhance the ability of the feature expression, too many convolutional layers will also lead to network gradient fading. Therefore, to ensure the feature expression ability of RSSOs while avoiding the network gradient fading caused by too many convolutional layers, the concept of a densely connected network (DenseNet) is adopted in the neck of YOLO-V5. DenseNet is composed of three core structures, namely DenseLayer (the most basic atomic unit of DenseNet), DenseBlock (the basic unit of DenseNet; also the core part of DenseNet) and Transition (the main function of which is to reduce the size and number of feature maps). It should be noted that the DenseBlock module can be regarded as stacking a certain number of DenseLayers, and DenseBlock is employed in this paper. The detailed concepts of DenseNet can be obtained from [44].

The network structure of the DenseNet block is shown in Figure 3, where $x_{i}$ means the feature map of the output, and $H_{0}\left(\cdot \right)$ and $H_{i}\left(\cdot \right)$ are the convolutional layer and the transport layer, respectively. Each layer can be connected to all other layers, which also means that each layer is able to receive all of the feature maps of the previous layer. With its unique structure, DenseNet can effectively alleviate the gradient attenuation of networks. Meanwhile, because each layer in the dense connected network will receive all of the feature mappings of the previous layer, the transmission of network features will be effectively enhanced, while the parameter volume of the network model will be also reduced to a certain extent.

It is worth noting that $H_{i}\left(\cdot \right),i=1,2,3,4$ are defined as: Conv(1×1×M)−BN−ReLU−Conv(3×3×M)−BN−ReLU. The input and output dimensions of the DenseBlock can be utilized to obtain the increment of each layer of feature mapping in “DenseBlock*i”. The advantages of the introduced DenseBlock over the C3 block of the neck are manifold. Firstly, under the same number of connection layers, DenseBlock can achieve better results and fewer parameters through the extreme use of features. Secondly, DenseBlock reduces the gradient disappearance caused by network depth to some extent. Finally, because the output feature map of each layer is the input of all previous layers, DenseBlock effectively enhances feature transfer while making more efficient use of features.

In view of the characteristics of RSI dataset, the number of small objects far exceeds that of medium objects and large objects. Based on the above factors, to obtain more abundant features and more accurate location information, the fourth dimension of the feature map, down-sampled four times, was selected in the proposed I-YOLO-V5 SOD framework as a new detection layer to be integrated into the network structure. The detailed algorithm structure diagram is shown in Figure 4.

## 4. Results

In this section, the RSOD dataset is introduced as the experiment dataset. The advanced OD methods, like YOLO-V5, YOLO-V7, YOLO-V8, etc., will be selected as competitors to compare with the proposed approach. The general experimental basis of this paper is as follows: GPU: NVIDIA GeForce RTX 2070. Environment: Python 3.8.8, PyTorch-2.1.1. Operating system: Ubuntu 20.04. The learning rate in this paper decreased from 0.001 to 0.00001 by cosine annealing.

### 4.1. Evaluation Metrics

To better quantify the binary classification method, the evaluation content consists of the following four categories: True Positive (TP), False Positive (FP), True Negative (TN), and False Negative (FN), which is exhibited in Table 1:

Based on the definition of confusion matrix in Table 1, the definition of precision and recall can be expressed as:(6)$$Precision=\frac{TP}{TP+FP}$$
(7)$$Recall=\frac{TP}{TP+FN}$$

Generally speaking, Precision can be utilized to measure the degree of false detection of OD, and Recall can be utilized to measure the degree of missed detection of OD. It is meaningless to only use Precision or Recall to measure the performance of OD, so it is necessary to consider Precision and Recall comprehensively. Based on this, the average accuracy (AP) and mean average accuracy (mAP), the most important evaluation indexes of OD, are introduced to quantify the detection accuracy of different types of objects, where the average accuracy can be expressed as:(8)$$AP_{i}=\int_{0}^{1}P_{i}\left(R_{i}\right)dR_{i}$$
where $P_{i}$ represents the precision of class i, and $R_{i}$ means the recall of class i. It is worth noting that $P_{i}\left(R_{i}\right)$ is expressed as a function, with $R_{i}$ as the independent variable and $P_{i}$ as the dependent variable.

The mAP can be defined as:(9)$$mAP=\frac{\sum_{i=1}^{c}AP_{i}}{c}$$
where *c* means the number of detected categories. It should be noted that AP and mAP, respectively, represent different detection performances. The former is to quantify the detection ability of a specific object, and the latter is to quantify the ability of all detection categories *c*.

Meanwhile, Frames Per Second (FPS) represents the number of frames of the image processed by the OD algorithm in one second, which plays an important role in evaluating the real-time performance of OD.

### 4.2. Experimental Process and Analysis

The RSOD dataset is selected as the experimental dataset to evaluate the validity of the proposed algorithm for RSOD. In this paper, the category size of the detected object is divided into three categories, namely the large object category (defined as the object pixel occupying more than 0.5% of the detected image), medium object category (defined as the object pixel occupying between 0.12% and 0.5% of the detected image), and small object category (defined as the object pixel accounting for less than 0.12% of the detected image). The objects marked in the sample are divided into four categories: aircraft, playground, oil tank, and overpass, where the aircraft and fuel tank objects are mostly small and medium-sized, and the scale of playgrounds and overpasses are large. The statistics of the experimental datasets are shown in Table 2.

In this paper, the evaluation indexes such as mAP, FPS, and miss rate are employed to measure the performance of the algorithm. Meanwhile, several advanced OD algorithms are competing with the proposed algorithm under the same evaluation indexes to verify the excellent performance of the proposed method in RSSO. The detailed comparison results of different algorithms are shown in Table 3. In addition, the comparison results that distinguish objects by size are shown in Table 4.

It can be clearly found from Table 3 that the mAP of the proposed algorithm (called I-YOLO-V5) is significantly better than other advanced algorithms, like YOLO-V5 and YOLO-V7, which also indicates that the improvement in this paper can enhance the detection ability of the algorithm to a certain extent. Compared with the original YOLO-V5 algorithm, although the I-YOLO-V5’s performance indexes are slightly different in terms of FPS, I-YOLO-V5 gained 15.4% in mAP compared to YOLO-V5, while I-YOLO-V5 only lost 5.6% in FPS compared to YOLO-V5. However, I-YOLO-V5 can still meet the real-time requirements of the algorithm in terms of FPS, which is better than YOLO-V5 in terms of comprehensive detection ability. It is worth noting that the AP value of the Faster RCNN on overpasses is 4% higher than that of I-YOLO-V5, the OD algorithm proposed in this paper. However, the detection speed of I-YOLO-V5 is nearly four times faster than that of Faster RCNN, compared with which the slight lowness of I-YOLO-V5 is negligible and the overall performance of I-YOLO-V5 in real-time application scenarios can be strongly verified. Meanwhile, it is reasonably seen from Figure 3 that I-YOLO-V5 can significantly improve the detection accuracy of small- and medium-sized objects, like aircraft and oil tanks, which can be seen from the AP of I-YOLO-V5. By studying Table 3, it is clearly recognized that, for overpasses with similar backgrounds, compared with several advanced OD algorithms like YOLO-V7 and YOLO-V8, I-YOLO-V5 shows better performance, which is reflected by the overpass’s AP value of I-YOLO-V5. From the values of FPS in Table 3, it can also be found that I-YOLO-V5 can meet the real-time performance requirements of RSO to a certain extent. And, it can be seen from Table 4 that I-YOLO-V5 has obviously enhanced the miss rate compared to other advanced algorithms, and can also significantly improve the detection accuracy of RSSOs under complex backgrounds.

It is worth noting from Table 3 that, compared to YOLO-V7 and YOLO-V8 and its derivatives, the mAP of I-YOLO-V5 does not show a significant advantage, but the proposed algorithm highlights better performance in the AP of aircraft and overpasses. It is not difficult to see from Table 3 and Table 4 that the current advanced OD algorithm has a very significant advantage in the detection performance of medium and large objects, but there is large room for improvement in the detection of small objects and the integration of background and objects. Based on the analysis of the above table, it is reasonably recognized that I-YOLO-V5 can meet the requirements of real-time detection, and has obvious effects in RSSO detection.

Figure 5 show some detection results of I-YOLO-V5 in different scenarios and under different conditions. From the comparison between Figure 5a and Figure 5b, it is not difficult to find that the complexity of the environment (such as the density of buildings) in RSSO detection will have a certain impact on the detection accuracy of the algorithm. Meanwhile, it can be found from Figure 5 that the detection accuracy of small- and medium-sized objects in Figure 5d is slightly better than that of small objects in Figure 5c, which further indicates that the light intensity has a certain degree of influence on the detection accuracy of small objects.

Figure 6 shows partial detection results for each category in the RSOD dataset, which not only finds common attributes, but also new findings that can find some of the same common attributes as discussed earlier, in addition to new findings. Compared to the detection of medium and large objects such as overpasses and playgrounds in RSIs, the detection accuracy of small objects such as aircraft and oil tanks is more obvious, which fully validates the excellent performance of I-YOLO-V5 in small OD of RSOs.

### 4.3. Ablation Experiments

To better verify the effectiveness of each of the combined modules involved in the proposed I-YOLO-V5 algorithm, a series of ablation experiments were conducted and are presented in this subsection. The related results are shown in Table 5 and Table 6. It is noted that the results of Table 5 are obtained by the OD with three detection layers, and those of Table 6 are obtained by the OD with four detection layers, i.e., the structure employed in this paper.

It is reasonably seen from Table 5 that, after the introduction of ‘Res2Unit’ and ‘DenseBlock’ in the algorithm structure, the mAP value increases from 76.62% to 77.84%, and the FPS value increases from 25.1 to 26.3%, which also verifies that the improvement of I-YOLO-V5 can effectively improve the detection performance and detection speed of the algorithm. It is worth noting from Table 5 that, compared with YOLO-V5, although the introduction of DenseBlock has increased mAP and FPS by 0.4% and 3.2%, respectively, the AP of aircraft has decreased by 0.36%, which also indicates that the detection performance of RSSOs cannot be effectively improved only by enhancing the transmission and reuse of feature maps to a certain extent. Table 6 shows the experimental data for the introduction of the fourth detection layer based on Table 5. It is clear from Table 6 that I-YOLO-V5 with the fourth detection layer has a more significant effect on improving the accuracy when detecting RSSOs such as aircraft, which also effectively proves that additional detection layer can improve the detection performance of RSSOs.

Based on the above experimental data analysis, it is not difficult to conclude that the introduction of each module in this paper is conducive to improving the detection accuracy of RSOs, especially the detection accuracy of RSSOs. Among the three improvements, the first item, ‘Res2Unit’, and the second item, ‘DenseBlock’, aim to improve the accuracy and detection speed of the algorithm, while the fourth detection layer introduced can significantly improve the detection performance of RSSOs, although part of the detection speed is sacrificed to a certain extent. In general, the proposed algorithm can meet the real-time detection requirements of RSSOs.

### 4.4. Comparison of Detection Effects

Figure 7 shows a partial representative comparison diagram of the detection performance between I-YOLO-V5 and the original YOLO-V5 algorithm, where a is the detection diagram of I-YOLO-V5, and b is the detection diagram of the original YOLO-V5 algorithm.

Figure 7 provides six groups of 12 images, which intuitively shows the detection effect of I-YOLO-V5 and YOLO-V5 algorithm. It can be clearly found from Figure 7a–d that YOLO-V5 cannot detect some objects effectively. The reason for the above phenomenon is that the volume of some RSSOs in the image is too small, and the color of the objects is close to the background color, which also verifies the excellent performance of I-YOLO-V5 in RSSOs from the side. Figure 7e,g show the detection comparison of the two algorithms under weak light conditions. It can be found that, although YOLO-V5 can detect most RSSOs, it is very easy to ignore small objects that are close to the background color. Compared with YOLO-V5, I-YOLO-V5 improves this phenomenon well. Figure 7g–l show the comparison of detection effects of medium and large objects under the two algorithms. It is not difficult to find that YOLO-V5 has the problem that objects cannot be effectively detected. The reason for the above phenomenon is that the detection object is too large and the feature is too close to the background color, resulting in missing detection of YOLO-V5, which is also the reason why the miss rate of YOLO-V5 in Table 4 is as high as 17.3%.

## 5. Conclusions

Aiming at the problems of small object sizes, high density, and colors being close to that of the background in RSIs, an improved YOLO-V5 algorithm is proposed and verified. To improve the multi-scale feature extraction performance of objects, a Res2Unit based on the idea of residual networks was proposed. Then, To improve the difficulty of feature extraction of small- and medium-sized objects in RSIs, the detection scales were increased from 3 to 4. In addition, to avoid gradient fading of the network, the DenseBlock structure is introduced into the structure of the algorithm. From the comparison of the results, it can be found that the performance of the proposed algorithm in RSSO detection is better than other advanced algorithms. Although the introduction of the fourth detection layer causes a slight decrease in the detection speed to some extent, it can still meet the real-time requirements of the algorithm. From the detection effect, it can be found that, compared with other classical modeling models, the proposed model is more suitable for the detection of RSSOs, and can meet the basic real-time detection requirements.

In future work, based on the proposed OD algorithm in this paper, how to speed up the detection inference without sacrificing detection accuracy and precision, and how to further integrate more effective visual attention mechanisms, as well as novel designs of high-resolution lightweight networks proposed in future, can be fully investigated.

## Figures and Tables

**Figure 1 sensors-24-04347-f001:**
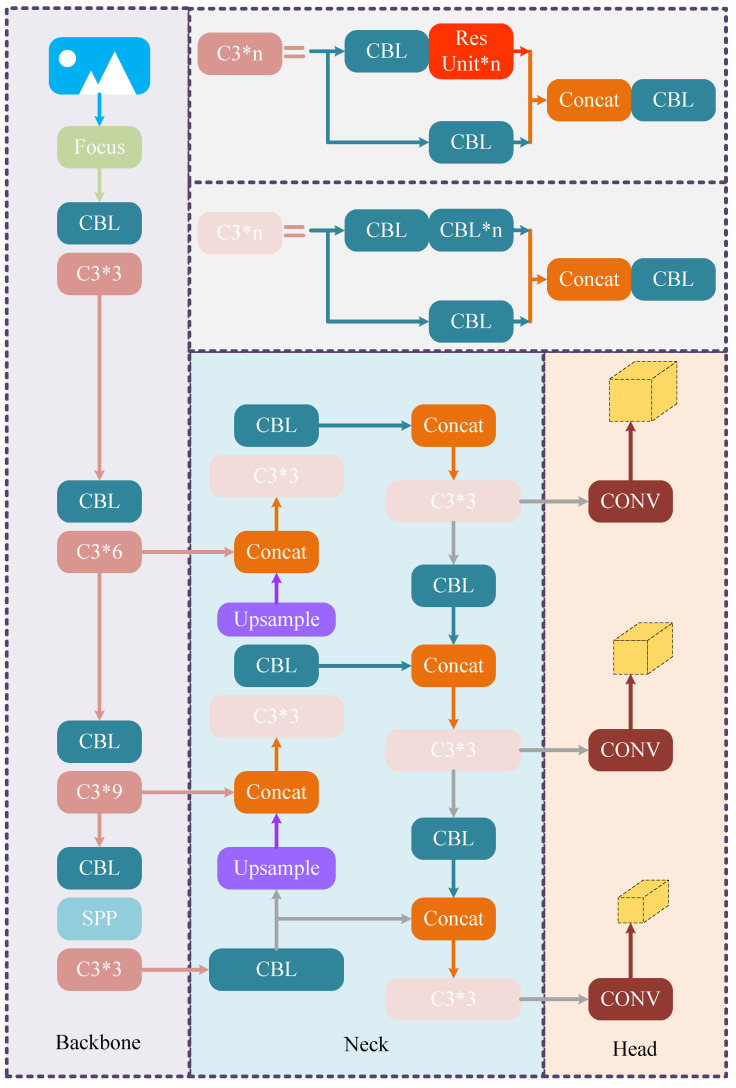
The architecture of YOLO-V5.

**Figure 2 sensors-24-04347-f002:**
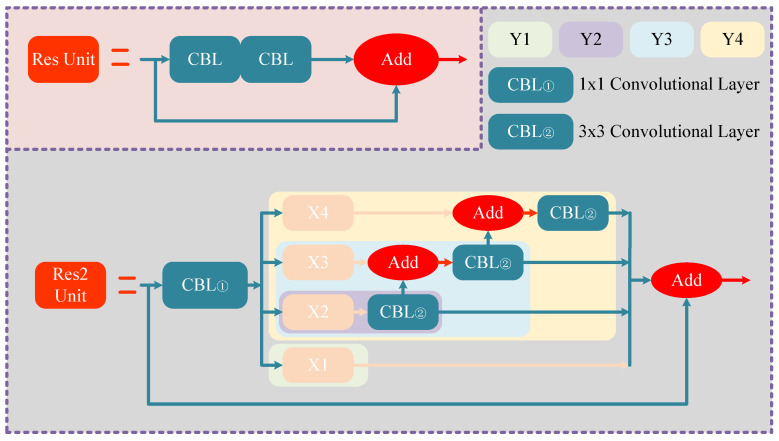
The structure of the Res2Unit.

**Figure 3 sensors-24-04347-f003:**
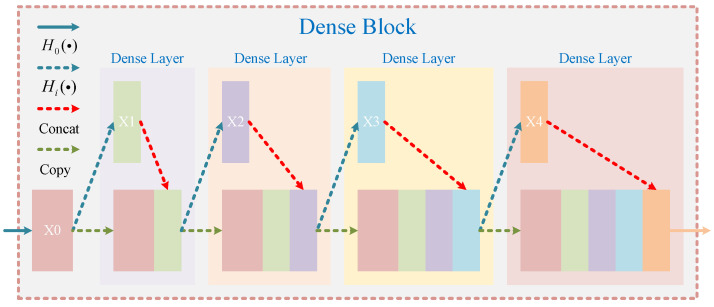
The structure of DenseBlock.

**Figure 4 sensors-24-04347-f004:**
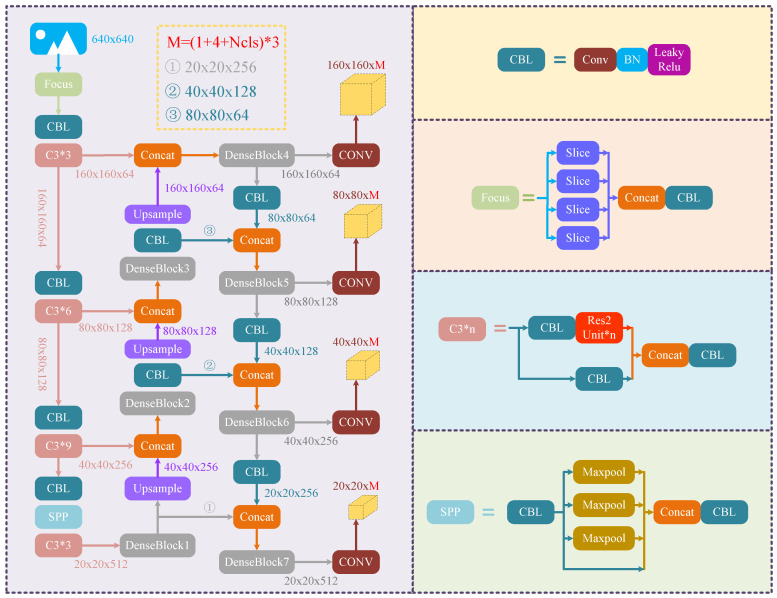
The structure of the proposed algorithm.

**Figure 5 sensors-24-04347-f005:**
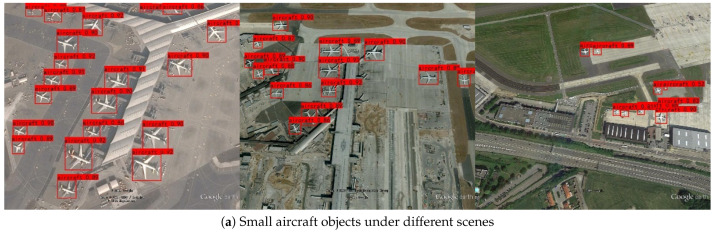
Tests under different conditions.

**Figure 6 sensors-24-04347-f006:**
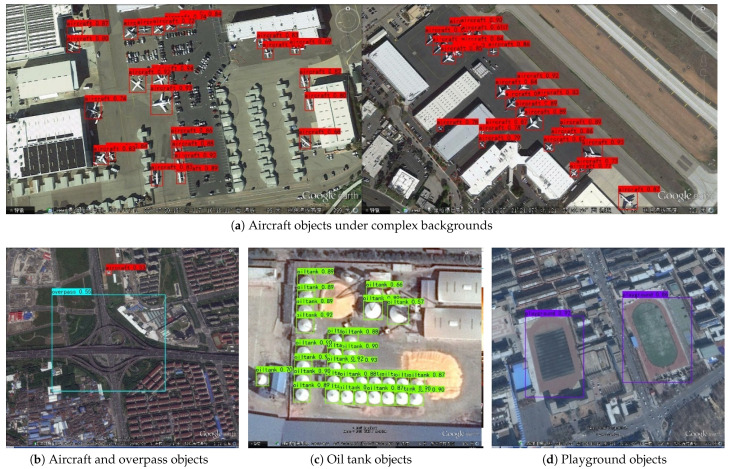
The detection results of I-YOLO-V5.

**Figure 7 sensors-24-04347-f007:**
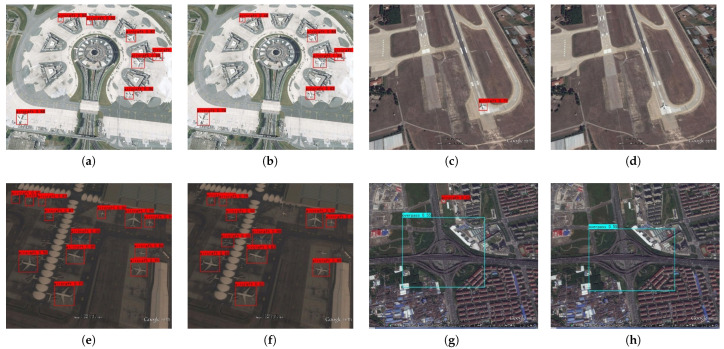
The comparison results of I-YOLO-V5 and YOLO-V5: (**a**,**c**,**e**,**g**,**i**,**k**) the detection results of I-YOLO-V5; (**b**,**d**,**f**,**h**,**j**,**l**) the detection results of YOLO-V5.

**Table 1 sensors-24-04347-t001:** The confusion matrix.

Actual	Predicted	Confusion Matrix
P	P	TP
N	P	FP
P	N	FN
N	N	TN

P: Positive; N: Negative

**Table 2 sensors-24-04347-t002:** The statistics of RSOD.

Dataset Name	Category	Number	Instances	Object Amount		
				S	M	L
RSOD	Aircraft	662	6250	4455	1192	603
	Oil tank	228	2153	981	926	246
	Overpass	212	221	0	0	221
	Playground	238	243	0	12	231

S: small; M: medium; L: large

**Table 3 sensors-24-04347-t003:** The comparison of different methods based on the RSOD dataset.

Algorithm	AP(%)					FPS
	Aircraft	Oil Tank	Overpass	Playground	mAP	
Faster RCNN	85.85	86.67	88.15	90.35	87.76	6.7
SSD	69.17	71.20	70.23	81.26	72.97	62.2
DSSD [45]	72.12	72.49	72.10	83.56	75.07	6.1
ESSD [46]	73.08	72.94	73.61	84.27	75.98	37.3
FFSSD [47]	72.95	73.24	73.17	84.08	75.86	38.2
YOLO	52.71	49.58	51.06	62.17	53.88	61.4
YOLO-V2	62.35	67.74	68.38	78.51	69.25	35.6
YOLO-V3	74.30	73.85	75.08	85.16	77.10	29.7
YOLO-V3 tiny	54.14	56.21	59.28	64.20	58.46	69.8
UAV-YOLO [48]	74.68	74.20	76.32	85.96	77.79	30.12
DC-SPP-YOLO [49]	73.16	73.52	74.82	84.82	76.58	33.5
YOLO-V5	78.51	76.38	74.64	76.96	76.62	25.1
YOLO-V7	87.27	89.12	82.78	87.31	86.62	28.7
YOLO-V7-CS [50]	85.88	94.41	80.40	91.73	88.11	25.9
YOLO-V7-ECA [51]	83.19	95.77	82.11	90.71	87.95	29.3
YOLO-V8	86.64	97.31	76.83	99.64	90.11	37.3
TSW-YOLO-v8n [52]	84.52	99.18	78.26	99.54	90.38	29.5
CS-YOLO-V8 [53]	85.63	97.77	80.41	99.52	90.83	16.8
I-YOLO-V5	89.91	90.34	84.55	88.92	88.43	23.7

**Table 4 sensors-24-04347-t004:** The comparison of different sizes in the RSOD dataset.

Algorithm	AP(%)			Miss Rate(%)
	S	M	L	
Faster RCNN	84.73	87.87	89.18	11.8
SSD	70.38	73.41	77.51	21.1
DSSD [45]	74.42	75.18	77.70	15.2
ESSD [46]	75.12	75.84	78.12	16.5
FFSSD [47]	72.62	74.78	82.56	18.2
YOLO	52.25	51.68	60.35	33.6
YOLO-V2	63.20	68.53	69.28	24.3
YOLO-V3	74.52	75.63	76.14	19.5
YOLO-V3 tiny	55.26	56.47	60.17	31.4
UAV-YOLO	75.45	75.15	76.85	17.1
DC-SPP-YOLO	75.41	74.67	76.41	15.9
YOLO-V5	79.21	72.32	81.34	17.3
YOLO-V7	87.31	88.34	88.34	13.7
YOLO-V7-CS [48]	85.34	95.87	92.49	12.9
YOLO-V7-ECA [49]	84.45	96.18	91.62	14.3
YOLO-V8	86.89	97.63	98.14	11.2
TSW-YOLO-v8n [50]	85.27	98.37	98.43	11.9
CS-YOLO-V8 [51]	86.75	98.17	98.84	10.7
I-YOLO-V5	89.13	90.25	92.37	9.2

S: small; M: medium; L: large

**Table 5 sensors-24-04347-t005:** The influence of improvement on RSSO detection with three detection layers.

Res2Unit	DenseBlock	AP(%)					FPS
		Aircraft	Oil Tank	Overpass	Playground	mAP(%)	
×	×	78.51	76.38	74.64	76.96	76.62	25.1
✓	×	78.72	76.15	75.44	77.23	76.89	25.4
×	✓	78.23	77.09	75.23	77.18	76.93	25.9
✓	✓	79.34	78.91	75.65	77.47	77.84	26.3

**Table 6 sensors-24-04347-t006:** The influence of improvement on RSSO detection with four detection layers.

Res2Unit	DenseBlock	AP(%)					FPS
		Aircraft	Oil Tank	Overpass	Playground	mAP(%)	
×	×	84.74	86.85	79.89	83.67	83.79	20.7
✓	×	84.96	87.05	80.22	83.98	84.05	21.2
×	✓	85.81	87.64	82.17	84.75	85.09	22.5
✓	✓	89.91	90.34	84.55	88.92	88.43	23.7

## Data Availability

The database used in this article is available at https://github.com/RSIA-LIESMARS-WHU/RSOD-Dataset-, accessed on 19 January 2017.

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
