# Peer review of "A Multi-Scale-Enhanced YOLO-V5 Model for Detecting Small Objects in Remote Sensing Image Information"

_sensors, 2024, doi:10.3390/s24134347_

Round 1
Reviewer 1 Report
Comments and Suggestions for Authors
The paper is well-written, follows a clear line of reasoning, and the idea is novel. I recommend acceptance for publication in Sensors when the following minor comments and questions are addressed:
The paragraph in lines 340-344 is not clear enough. Please reword it.
Why does the proposed OD algorithm perform lower for identifying the Overpass category (AP = 84.55%) than the Faster RCNN?
Could you add your recommendations as future work to improve the performance of the OD algorithm proposed even more?
Please add the reference to the RSOD dataset
Comments on the Quality of English LanguageThe paragraph in lines 340-344 is not clear enough. Please reword it.
Reviewer 2 Report
Comments and Suggestions for Authors
1. A major concern for 3.2. You aim to improve YOLO for small object detection tasks. However, in this key part of whole paper, I do not see any relationship between structure improvement and small object detection task.
2. From the eq.5, I noted these four sub-feature maps are calculated in parallel, Whether this design affect the detection efficiency? Need the ablation study.
3. Too many details for DenseNet in Lines 229-246. It is well known in the community, please just give a brief introduction. The same thing lines 260-279, these metrics are basic in the community, no more details.
4. Fig.1 is not self-contained, you need to supplement some details in the title. In this version, it is hard to understand ‘focus’ ‘cbl’ et.al for readers who are not familiar with this field.
5. About Sign. For eq.1 IoU is wider used compared to IOu. For eq. 2 ${\ell }$ rather than loss
6. Some mistake.
* miss ‘.’ For line 228
* Table 5 6,No Yes please not they are case-sensitive
Comments on the Quality of English LanguageNeed Improve
Reviewer 3 Report
Comments and Suggestions for Authors
I would like to thank the authors for their valuable efforts.
When reviewing your work, several questions and comments have arisen.
In the paper, the numbering of references is not in order, but alphabetically, which is immediately noticeable. Perhaps the numbering should be done sequentially.
This link https://github.com/RSIA-2039920LIESMARS-WHU/RSOD-Dataset- does not work.
AP (%) YOLO-7 and YOLO-8 are almost identical to I-YOLO-V5. Don't you think that the newer versions of the algorithm already have the improvements you suggest in your paper? Why was the 5th version of the algorithm initially taken, and not the 8th?
Comments on the Quality of English LanguageText is clear to read; however. Some minor grammar corrections should be considered.
Round 2
Reviewer 2 Report
Comments and Suggestions for Authors
The authors have addressed all my concerns. I vote for publication.